# Behavioral Characteristics of Largefin Longbarbel Catfish *Hemibagrus macropterus*: Effects of Sex and Body Size on Aggression and Shelter Selection

**DOI:** 10.3390/ani15091192

**Published:** 2025-04-22

**Authors:** Xiaoli Li, Yongjiu Zhu, Siqi Chen, Tingbing Zhu, Xingbing Wu, Xuemei Li

**Affiliations:** Key Laboratory of Freshwater Biodiversity Conservation, Ministry of Agriculture and Rural Affairs, Yangtze River Fisheries Research Institute, Chinese Academy of Fishery Sciences, Wuhan 430223, China; lxl@yfi.ac.cn (X.L.); zhuyj@yfi.ac.cn (Y.Z.); csq170912040@163.com (S.C.); zhutb@yfi.ac.cn (T.Z.)

**Keywords:** *Hemibagrus macropterus*, social hierarchy, nocturnal aggression, habitat preference, intraspecific competition

## Abstract

This study explored the aggressive behavior of *Hemibagrus macropterus* (Bleeker, 1870), a commercially valuable fish, to understand how size, sex, and shelter availability influence their interactions. By observing fish of different size combinations (large vs. large, large vs. small, small vs. small), sex combinations (male vs. male, male vs. female, female vs. female), and shelter types (tiles, pebbles, grass) under varying group densities (1, 3, and 6 fish), we found that *H. macropterus* is more active and aggressive at night, establishing a social hierarchy with dominant and subordinate individuals. Larger fish and females showed higher aggression, but the presence of shelters, especially tiles and pebbles, significantly reduced aggressive interactions. Fish preferred tile and pebble shelters, with no cohabitation in these, while aquatic plants allowed two fish to share space. Aggression was higher in larger groups, suggesting that managing group size and providing adequate shelters can reduce conflicts. These findings highlight the importance of nocturnal feeding, size-segregated rearing, and sufficient shelters in aquaculture to minimize aggression, improve fish welfare, and enhance cultivation efficiency. This research provides practical insights for optimizing the culture of *H. macropterus* and contributes to understanding the behavior of similar fish species.

## 1. Introduction

Animal aggression is an adaptive behavioral characteristic that serves as a primary driver for territory establishment and the formation of social hierarchies [1]. It also plays a crucial role in securing limited resources in nature, including food, mates, and nesting sites [2,3]. In aquatic animals, aggressive behaviors are also present, particularly in crustaceans as well as in Siluriformes fish species [4,5,6,7].

Factors influencing aggressive behaviors in aquatic animals can be categorized into three main groups: (1) environmental factors, including temperature, light, and salinity; (2) individual factors, such as size, sex, and social experience; (3) genetic factors [7]. In fishery production practices, measures are implemented to reduce aggressive behaviors based on these influencing factors. These measures include environmental regulation, density control, size heterogeneity, and improved feeding practices. For instance, reducing fish density has been effective in decreasing aggressive behaviors in ornamental fish species such as Swordtail (*Xiphophorus helleri*)*,* Angelfish (*Pterophyllum scalare*), and Tiger Barb (*Puntigrus tetrazona*) [8,9,10]. Su et al. found that the intensity, duration, and frequency of fights among Japanese Blue Crab (*Portunus trituberculatus*) increase with temperature [11]. Therefore, during high-temperature periods in the *P. trituberculatus*’s cultivation, setting up shelters can help reduce aggressive behaviors. In fishery practices, enhancing the defensive capabilities of cultured animals, alleviating predation stress, avoiding aggressive behaviors, and ultimately improving survival rates and increasing aquaculture benefits can be achieved by setting up shelters in the cultivation environment [12]. For instance, in the cultivation of Crayfish (*Procambarus clarkii*), especially in intensive indoor facilities, considering square-shaped cultivation structures and appropriately adding bricks or similar tools to increase corner numbers according to the cultivation density and quantity of *P. clarkii* can improve their habitat or shelter conditions, thereby promoting their growth. Additionally, to reduce the complexity of feeding operations and the cannibalistic behavior of cultured fish that may arise from growth heterogeneity within the feeding group, size grading has become a common practice in aquaculture [13].

Siluriformes fishes are widely distributed, with many species extensively cultured. Reports have indicated that some Siluriformes species exhibit aggressive behaviors [14,15]. For example, subterranean Copionodontinae catfish (*Siluriformes: Trichomycteridae*) *Copionodontinae*, also known as the Brazilian cave catfish, is described as one of the more aggressive species among subterranean fish, characterized by high attack frequency and complex attack composition [16]. Research indicates that the dominant behaviors of Siluriformes fish are correlated with factors such as body size, physical strength, and territoriality. Dominant individuals may maintain their status through sustained aggression [17]. *H. macropterus*, a medium-sized demersal fish, belongs to the class Teleostei, Siluriformes, and Bagridae, as well as the genus *Mystus*. It is naturally distributed in the Pearl River, Xiangjiang River, Ganjiang River, and Yangtze River systems, where they commonly inhabit rapids and rocky gravel environments [18]. It is also a commercially valuable species known for its rapid growth and large size, with high contents of protein, fat, essential amino acids, and savory amino acids, thereby offering high nutritional value [19]. Over the course of research on the artificial breeding and culture of *H. macropterus*, it was observed that these fish exhibit aggressive behaviors, resulting in wounds, ulcers, and even fatalities [20]. Understanding the behavioral traits and influencing factors of *H. macropterus* is essential to mitigate, or ideally eliminate, their aggressive interactions.

Thus, this study investigated the behavioral characteristics of *H. macropterus* based on three factors: body size, sex, and shelter type. The specific details are as follows: ① The impact of size specifications (large vs. large (LL), large vs. small (LS), small vs. small (SS)) on aggressive behavior. ② The influence of sex (male vs. male (MM), male vs. female (FM), female vs. female (FF)) on aggressive behavior. ③ By constructing three types of caves (tiles, pebbles, and grass) in indoor aquaculture tanks, and setting up groups with different fish numbers—free-choice group (1 fish), competition-absent group (3 fish), and competitive group (6 fish)— the study analyzed the choices of shelters (tiles, pebbles, and grass) by *H. macropterus* under various conditions (free-choice, competition-absent, and competitive). The aim was to provide a scientific basis for the pond culture technology of *H. macropterus*, and to accumulate foundational data for behavioral studies of Siluriformes fish represented by *H. macropterus*.

## 2. Materials and Methods

### 2.1. Experimental Materials

The *H. macropterus* specimens used in this study were obtained from the Endemic Fish Breeding and Cultivation Base of the Yangtze River Fisheries Research Institute, Chinese Academy of Fishery Sciences, where healthy individuals with uniform size and no physical deformities were selected and acclimated in rectangular indoor aquaculture tanks (4 m × 5 m) under controlled environmental conditions. Detailed biometric parameters were recorded (Table 1), with wet body weight measured using a precision electronic balance (Shanghai Sunny Hengping Scientific Instrument Co., Ltd., Shanghai, China, accuracy ±0.01 g) after surface moisture removal and total length determined with a digital caliper (accuracy ±0.1 mm) on anesthetized specimens. Sex was differentiated morphologically, with males exhibiting pointed urogenital papillae and streamlined abdomens while females showed rounded papillae and broader abdomens, a method validated with 100% accuracy through gonadal examination of a 30% subsample. Fish were fed twice daily with commercial extruded feed (Hubei Chia Tai Feed Co., Ltd., Wuhan, China; crude protein ≥ 45%, crude fat ≥ 8%, moisture ≤ 12%, ash ≤ 16%) at 07:00 and live bait fish (*Cirrhinus molitorella*, ~5 cm) at 20:00, while strict welfare protocols were maintained including twice-daily health inspections, immediate isolation of injured individuals (>2 mm lesions; <1% occurrence), and a study termination threshold (>10% moderate injuries; not triggered). All procedures complied with institutional animal care guidelines and ethical standards.

The experiments were conducted in circular indoor cement tanks (2.4 m diameter × 1 m depth) with smooth ceramic-tiled surfaces. Each tank was maintained at a water depth of 0.6 m and divided into four equal sectors using 5 cm wide yellow adhesive tape markers. The system utilized filtered and pre-aerated pond water (48 h pretreatment period) with the following stable water quality parameters: temperature 18.0 ± 0.5 °C, dissolved oxygen 8.0 ± 0.5 mg/L, and pH 7.9 ± 0.2. A complete water exchange was performed between experimental trials, with no water changes or supplemental aeration during testing periods. Environmental conditions included a controlled photoperiod (11L:13D, automated digital timer) with surface light intensity ranging 0–500 lux (LI-250A light meter, LI-COR Biosciences, Lincoln, NE, USA). The experimental area maintained acoustic isolation, with no extraneous anthropogenic disturbances except for scheduled feeding activities.

### 2.2. Experimental Design and Methods

#### 2.2.1. Body Size Effects Experiment

Two tile caves were constructed in one of the aforementioned circular tanks, with tile dimensions of 40 cm (length) × 24 cm (width) × 12 cm (height), featuring a semicircular cross-section and openings at both ends. The tile caves were placed at the center of the corresponding areas within the circular tank, with the openings facing the center of the tank. The experimental fish were divided into three groups—large vs. small, small vs. small, and large vs. large—with two fish in each group. The fish were placed in the circular tank for a continuous observation period of 72 h, after which one tile cave was removed. Following another 72 h observation period, the remaining tile cave was also removed. Each experimental group had three replicates under identical conditions, and each fish was used for only one experiment. The fish selected for the size experiment were chosen randomly without consideration of sex.

#### 2.2.2. Sex Effects Experiment

Tile caves were constructed in the second circular experimental tank according to the method described in Section 2.2.1. The experimental fish were divided into three groups—female vs. male, male vs. male, and female vs. female—with two fish in each group. The fish were placed in the circular tank for a continuous observation period of 72 h, after which one tile cave was removed, and the remaining tile cave was also removed another 72 h later. Each experimental group had three replicates, and each fish was used for only one experiment. The sizes of the fish in the sex experiment were generally uniform.

#### 2.2.3. Shelter Selection Experiment

In the third circular tank, four zones were established: a tile cave area, a pebble cave area, an aquatic plant cave area, and an open area (control). In the tile cave area, the specifications were the same as those in Section 2.2.1. In the pebble cave area, pebbles sized 5–10 cm were selected to construct caves with an area of 40 cm × 24 cm; tiles and bricks were used for support to prevent the pebbles from sliding. In the aquatic plant cave area, Elodea with a length of 30–50 cm was collected from the aquaculture pond and bundled together with nylon rope, with bricks suspended from the bottom for stabilization. The open area was left devoid of any shelter. The caves were all positioned at the center of their respective zones, with openings oriented toward the center of the experimental tank.

The experimental setup in the tank containing the three types of shelters was divided into three groups based on the number of fish—free-choice group (1 fish), competition-absent group (3 fish), and competitive group (6 fish)—aiming to investigate the behavioral characteristics of *H. macropterus* and their selection of shelters under different conditions. Each group had three replicates, and each fish was used for only one experiment (Figure 1).

### 2.3. Data Collection

The entire experimental process was observed using a video recording. The observation system comprised a network camera with night vision infrared capabilities (Hikvision, DS-2CD864, Hangzhou, China) and a monitor (Hikvision, DS-8832N-R8, China). The network camera was fixed 2 m above the water surface of the experimental tank. Following the introduction of the experimental fish into the tank, recording commenced immediately, with each recording session lasting for 72 h. During the observation period, feeding management for *H. macropterus* was consistent with that of the acclimatization phase.

### 2.4. Behavioral Quantification and Data Analysis

The behaviors of *H. macropterus* were analyzed frame by frame using motion analysis software to identify and detail all components of the fish’s behavior (Table 1). Based on the classification methods of animal behavior according to references [16,21], the behaviors of the experimental fish were categorized into daily behavior, territorial behavior, aggressive behavior, and others. Among them, ‘aggressive behavior’ encompasses all agonistic interactions; ‘attack frequency’ specifically quantifies physical contacts per observation period. A continuous video record for 24 h (from 10:00 to 10:00) was selected from each experimental group to quantify and statistically analyze the following parameters: the number of fish distributed across different areas, the frequency of entering and exiting shelters, the duration of stay in various shelters, and the frequency of aggressive interactions. Prior to analysis, all datasets were assessed for normality using Shapiro–Wilk test (*n* < 50) and for variance homogeneity using Levene’s test. Parametric data (*p* > 0.05 for both tests) were analyzed by one-way ANOVA with Tukey’s post-hoc test in Origin 2019, while non-parametric data (*p* ≤ 0.05) underwent Kruskal–Wallis test with Dunn’s correction. Results are expressed as mean ± SD, with α = 0.05 defining statistical significance.

## 3. Results

### 3.1. Behavioral Characteristics of H. macropterus

*H. macropterus* exhibited distinct behavioral patterns characterized by a crepuscular activity cycle; they sought refuge in caves during the day and were highly active at night, with elevated aggression rates when maintained in groups. Specific behaviors and their characteristics are summarized in Table 1. The group housed with multiple individuals displayed all the behaviors listed in Table 2, while the solitary fish exhibited only daily behaviors.

#### 3.1.1. Daily Behavior

Daily behaviors were observed in both the solitary fish group and the group with multiple fish coexisting. After introduction into the tank, *H. macropterus* quickly swam towards the tile cave and concealed itself within. After approximately 2 to 3 h, it displayed exploratory behaviors, characterized by repeatedly protruding its head from the cave [mean of (10 ± 2) times, duration ranging from 50 s to 2 min 56 s]. Subsequently, the fish would swim out of the cave and return swiftly [mean of (6 ± 2) times, with an average duration of 1 min 20 s to 2 min 59 s, covering an average distance of (38 ± 10) cm]. The fish then swam upward to the surface before returning, followed by emerging from the cave, crossing the boundary, and again swimming out of the cave. After swimming longer distances along the tank walls (crossing two boundaries), it would return without re-entering the cave, indicating the end of exploratory behavior, with a total average duration of (13 ± 2) minutes. This exploratory behavior was only observed during the initial emergence from the cave after being placed in the tank, subsequently giving way to exploration, patrol, and guarding behaviors. These behaviors occurred in an overlapping manner without a clear sequence or specific duration. The average duration of exploratory behavior in the multiple fish group was approximately (39 ± 5) minutes, about three times that of the solitary fish.

#### 3.1.2. Territorial Behavior

*H. macropterus* demonstrated a significant sense of territoriality, specifically manifested in two ways: (1) a preference for remaining in a specific location, whether in shelters or open areas; (2) exhibiting guarding behavior, responding to the presence of other fish by displaying aggressive eviction actions, and then returning to the original location (either an open space or within the cave) to remain stationary or continue patrolling until ensuring that no intruders were present.

#### 3.1.3. Aggressive Behavior

Aggressive behaviors were observed when multiple *H. macropterus* coexisted, involving pursuits, counter-pursuits, collisions, bites, evasion, and mutual avoidance (Table 2). These aggressive interactions occurred when two individuals encountered each other and their interactions were not significantly related to feeding or habitat selection.

Based on these aggressive behaviors, a social hierarchy was established, distinguishing between “dominant individuals” and “submissive individuals”. This hierarchy was reflected in the preferential selection of shelters by dominant fish, who favored tile caves and pebble caves, constantly alternating between them. Conversely, submissive individuals were forced to remain in grassy areas or open spaces. Attempts by these submissive fish to enter tile or pebble caves were met with eviction by resident dominant fish, often leading them to seek refuge in grassy caves. However, subsequent conflicts would force the submissive individuals back into open areas. The social hierarchy was further evidenced by the “intimidation” exerted by dominant individuals on submissive ones. When a dominant fish left a tile cave, a submissive individual would approach but only enter after multiple observations and confirmations of the absence of the dominant resident. Sometimes, fish in open areas would approach tile caves, but upon spotting a dominant individual inside, they would swiftly flee.

Consistent with the nocturnal activity patterns of *H. macropterus*, aggressive behaviors also exhibited a significant diurnal rhythm. The frequency of aggression was significantly higher during the night compared to daytime (*p* < 0.05). Specifically, in the competition-absent group, the nighttime aggression frequency was 22.85 ± 3.30 times higher than during the day, while in the competitive group, it was 2.28 ± 0.30 times higher. Additionally, aggression frequency increased with encounter frequency, with the daytime aggression frequency in the competitive group being 18.00 ± 1.85 times higher than in the competition-absent group, and the nighttime aggression frequency being approximately 2.00 ± 0.15 times higher.

### 3.2. The Impact of Size on Aggressive Behavior of H. macropterus

The size experiment also demonstrated a significant tendency for *H. macropterus* to be nocturnal, with attack frequency during the night being significantly higher than during the day (excluding the small–small group). Additionally, the frequency of attacks in environments without caves was significantly higher than in those with caves. Regardless of the presence of caves, the attack frequency among large-size *H. macropterus* was significantly greater than that observed in the large–small and small–small groups (*p <* 0.05). In environments without caves and with limited cave availability (no caves and one cave), the large–small group exhibited a slightly higher attack frequency compared to the small–small group, though the difference was not statistically significant. Furthermore, in scenarios involving cave competition, there was no noticeable aggressive behavior between the large–small and small–small groups. Video recordings indicated that within similar size groups, dominant and subordinate individuals emerged, with the dominant individuals consistently attacking the subordinates, who predominantly opted to evade. While caves provided a temporary refuge, dominant fish would also follow subordinates into the caves (Figure 2).

### 3.3. The Impact of Sex on the Aggressive Behavior of H. macropterus

There are significant sex characteristics in the aggressive behavior of *H. macropterus*. The female–female and female–male groups exhibited significantly higher attack frequencies compared to the male–male experimental group (*p <* 0.05). The male–male group engaged in fewer attacks, primarily maintaining a defensive stance and confrontation without moving away from each other. They remained vigilant in their confrontation, showing no overt pursuit behavior (Figure 3).

### 3.4. Selection of Shelter by H. macropterus

In free-choice conditions, *H. macropterus* consistently selected tile caves during the day and remained inside. At night, they also showed a preference for tile caves, with the frequencies of entering and exiting tile caves, pebble caves, and grass caves being (7.78 ± 0.80) times/h, (3.62 ± 0.55) times/h, and (1.15 ± 0.08) times/h, respectively. The average durations of stay were (11.38 ± 1.82) min/h, (6.53 ± 0.72) min/h, and 0 min/h, respectively. As illustrated in Figure 4, there were significant differences in the selection of the three types of shelters by the free-choice group of *H. macropterus* (*p <* 0.05), with preferences ranked from highest to lowest as follows: tile, pebble, and grass.

In the competition-absent group, the frequencies of entering and exiting the three types of shelters and the durations of stay are illustrated in Figure 5. The frequencies and durations of stay for fish moving in and out of tile caves and pebble caves were significantly lower than those for grass caves (*p* < 0.05). This can be attributed to the preference of “dominant individuals” for pebble caves and tile caves, as they continuously rotated between the two. In contrast, “subordinate individuals” were restricted to staying in grass or open areas. During this time, fish that remained in open areas intermittently attempted to enter tile caves or pebble caves, but were driven away by the resident individuals, forcing them to retreat to the grass caves or remain in open areas.

The frequencies of entering and exiting the three types of shelters and the durations of stay for *H. macropterus* in the competitive group are shown in Figure 6. In the presence of competition, the frequencies of entering and exiting pebble caves, tile caves, and grass caves during the day were (12.50 ± 1.30) times/h, (12.50 ± 1.10) times/h, and (11.00 ± 1.20) times/h, respectively. The durations of stay were (5.55 ± 0.43) h, (4.39 ± 0.55) h, and (10.40 ± 1.20) h, respectively. There were no significant differences in the frequencies of entry and exit among the three types of shelters (*p* > 0.05). However, the duration of stay in grass caves was approximately equal to the combined durations in tile caves and pebble caves. This is attributed to the fact that two or more fish cannot coexist in tile and pebble caves, while two fish can share the grass caves.

As shown in Figure 4, Figure 5 and Figure 6, all groups exhibited a significant diurnal activity pattern in *H. macropterus*, characterized by daytime rest and nocturnal activity. When multiple fish coexisted, the frequencies of entering and exiting various caves during the day in the competitive group were approximately 10 times higher than those in the competition-absent group, and the durations of stay were about 3 times longer. During the night, the frequencies of entering and exiting caves in the competitive group were about 3 times higher than those in the competition-absent group, significantly higher than the proportion of fish (2 times), indicating a significant burrowing habit in *H. macropterus*. Given the insufficient number of shelters in the competitive group, all fish attempted to enter the shelters. Additionally, there was a longer duration of fish staying in various shelters at night among the competitive group compared to the competition-absent group, where almost no such behavior was observed. This suggests that during nocturnal activity, fish engage in aggressive encounters due to encounters, and the shelters provide them with places to hide.

From the analysis of the number distribution of experimental fish in each area and the selectivity index (Figure 7), the competition-absent group showed the highest selectivity index for grass caves, followed by tile caves and pebble caves, as “dominant individuals” rotated their residence between pebble caves and tile caves. In the competitive group, the highest number of fish chose the tile cave area, likely because some fish positioned themselves outside the tile caves, waiting for an opportunity to enter; the second most chosen area was the grass cave, where two fish could coexist, followed by pebble caves and open areas.

## 4. Discussion

### 4.1. Diurnal Rest and Nocturnal Activity, and Aggressive Behavior in H. macropterus

In this study, all groups of *H. macropterus* exhibited significant diurnal rest and nocturnal activity, consistent with the behavioral characteristics of other cave-dwelling catfish species [22]. This diurnal rest and nocturnal activity behavior not only increases the activity range but also avoids adverse effects, which is the result of natural selection during the long-term evolution process [23]. Research shows that benthic fish generally exhibit negative phototaxis [24,25]; this behavior is because higher illumination during the day makes the body more visually conspicuous, making them more likely to be detected. Therefore, they prefer to hide in places with shelter to gain a sense of security [21]. In aquaculture practices, this diurnal rest and nocturnal activity behavior can be utilized by adopting a night feeding mode to improve farming efficiency.

*H. macropterus*, as a benthic species, exhibits typical aggressive behaviors. This is because the competition among benthic fish species is more intense in underground environments than in surface environments [26]. Several factors are believed to be related to mutual aggression, including territorial defense, food competition, and mating partners (defense of mating territory) [16]. In this experiment, there was abundant food with no competition among *H. macropterus* observed, so the main reason for aggression was related to territorial defense. In aquatic animals, crayfish determine their hierarchy through fighting [27]. In some cichlid species (*Geophagus brasiliensis*), subordinates usually have dark stripes around their eyes after fighting, while dominant fish are usually lighter in color [28]. In *P. clarkii*, 71% of the shelters were occupied by dominants, and 7% were occupied by subordinates through fighting [29]. In this experiment, *H. macropterus* established a significant social hierarchy through aggressive behaviors, distinguishing between “dominant individuals” and “subordinate individuals”. The “dominant individuals” intimidated the “subordinate individuals”, reducing persistent aggressive behavior. This indicates that to some extent, the above social hierarchy can effectively prevent related injuries caused by persistent aggressive behavior. The relationship between the sex, size of *H. macropterus*, and the establishment of social hierarchy also needs further study.

### 4.2. Influence of Individual Size on Aggressive Behavior

In certain fish populations, hierarchical structures arise due to differences in body size. In groups consisting of individuals of varying sizes, attacks are more frequently observed among larger individuals of similar size than between individuals of differing sizes [30,31]. These effects can be explained by the fact that some fish, particularly territorial species, utilize cues related to the relative sizes of their competitors to assess their fighting ability. Individuals of similar size are more likely to be perceived as potential rivals, leading to an increase in aggressive interactions [31,32,33]. According to the sequential assessment model proposed by Enquist and Leimar [32], opponents are assumed to use the relative size of their competitors as a cue to evaluate their fighting capabilities as conflicts unfold [34]. When the size asymmetry is minimal, conflicts are more likely to escalate [33]. Thus, in size-mixed groups, significant size differences can quickly establish hierarchies, leading smaller fish to become timid and cautious to avoid attacks or predation by larger fish [35]. In contrast, larger fish of similar size may escalate conflicts further to assess their relative fighting abilities and establish a clear dominance hierarchy. This has been corroborated in studies of *Sebastes schlegelii*, which demonstrated that aggressive interactions predominantly occur among individuals of similar size rather than between those of differing sizes [13]. In this study, a similar trend was observed, whereby aggression was more intense among larger individuals. The tendency for aggression may diminish between larger and smaller fish or among smaller fish, as smaller individuals are likely perceived as relatively weaker competitors, making them more inclined to evade competition and reduce conflict. Such behaviors may be influenced by the behavioral characteristics of fish, such as territoriality. Therefore, in aquaculture, classifying fish by size could mitigate the suppressive effects of social hierarchies on the growth of smaller individuals [13].

### 4.3. Influence of Sex on Aggressive Behavior

In addition to body size, another individual trait that may influence aggressive behavior is the sex of the individual. The majority of studies on competitive dynamics have focused on competition between males, possibly due to the assumption that males engage in intense competition over a limited number of females [36]. While males typically engage in aggressive interactions with other males for mating opportunities, it is well known that females also exhibit aggression towards other females [37] and males [38]. Generally, males and females differ in their motivations, rules, and rewards of competition [38,39,40]. However, other studies on sex-specific competition have shown no relationship between sex and the duration, intensity, or likelihood of winning fights [41], suggesting that sex differences are not universally present.

In this study, the FF and FM groups exhibited significantly higher aggression levels compared to the male–male experimental group, indicating that females are more aggressive. This may be related to reproductive behaviors and territorial defense: In some species, females may also exhibit aggressive behaviors to defend their territories or resources, ensuring successful reproduction and offspring survival during the breeding season [42]. Additionally, females may display aggressive behaviors to protect their reproductive resources, such as eggs or fry, to ensure the survival of their offspring. For example, female sticklebacks may show aggression during the breeding season to protect their nests and eggs [43]. Such behaviors can be seen as a survival strategy, enhancing reproductive success and offspring survival rates. This aligns with the parental investment theory and resource defense hypotheses, where females may exhibit heightened aggression to protect feeding territories critical for oocyte development [44,45]. Furthermore, as this experiment was conducted during the non-breeding season of the species, competition might occur without reproductive considerations or motivations. Further research could test for sex effects during the breeding season to verify whether these effects are more pronounced during that period, thereby gaining a better understanding of this possibility. Additionally, future studies could investigate other sex-related traits, such as relative gonad weight as a predictor of competition outcomes [46], to further support the role of reproductive considerations in competitive dynamics.

The above results indicate that for this species, both individual size and sex are critical factors in determining competitive dynamics, and both aspects need to be considered in future breeding and aquaculture processes.

### 4.4. Shelter Selection by H. macropterus

The selection of living environments is one of the territorial behaviors exhibited by aquatic animals. Research on these territorial behaviors helps in the construction of suitable habitats in aquaculture production and improves the survival rate of fry. In this study, the preference of *H. macropterus* for different shelters, in descending order, was tile, pebble, grass, and open spaces. Field investigations have shown that *H. macropterus* often inhabits the undersides of slabs or stone cavities in shallow upper river sections, which is consistent with the results of this experiment. The main factors influencing animal survival selections are foraging, reproduction, and predator avoidance. In the wild, *H. macropterus* primarily feeds on benthic invertebrates such as aquatic insects, snails, clams, shrimp, and crabs [47]. These organisms are often densely distributed at the bottom of water bodies and in crevices between stones, which may explain why *H. macropterus* prefers pebble and tile caves.

In this study, only one fish could inhabit tile and pebble caves at a time, while two fish could cohabit in grass caves. This may be related to the strong territorial consciousness of *H. macropterus*. Animal territoriality and behavior typically involve the occupation and protection of a specific space [1]. Occupying space provides access to survival resources and assists in courtship, mating, reproduction, and offspring rearing [48,49]. Territorial behavior generally exhibits strong exclusivity. For example, male *Balistes capriscus* actively protect the area around their nests from other male *B. capriscus* and other fish species before fertilization, and they continue to defend the territory and chase away other fish after fertilization [50]. The territorial consciousness of *H. macropterus* is demonstrated by its preference to stay in a fixed position and exhibit guarding or chasing behaviors when other fish pass by.

Research has shown that shelters, as essential resources for aquatic animals, can reduce the intensity of intraspecific fighting and increase survival rates [29,51]. In this experiment, *H. macropterus* showed a greater preference for tile and pebble caves. However, these caves could not be cohabited by multiple fish. Grass caves, being more open shelters, may obstruct the fish’s view with grass roots in the center, reducing the chance of fish encountering each other and thus minimizing aggressive behavior, allowing cohabitation. In the practice of *H. macropterus* aquaculture, it may be appropriate to construct open-type shelters that provide both refuge and more foraging opportunities.

### 4.5. Relationship Between the Number of Shelters and Attack Frequency

In this experiment, the attack frequency of the competitive group (6 fish) during the day was 18 times/h, while that of the competition-absent group (3 fish) during the night was 22.85 times/h. The former was slightly lower than the latter, with no significant difference. The reason was that 3–4 fish in the competitive group hid in the shelters during the day, so the number of fish in open spaces during the day in the competitive group was numerically close to the number of fish in the competition-absent group at night, resulting in similar encounter and attack frequencies. According to probability theory calculations, in the absence of shelters at night, the theoretical encounter frequency of *H. macropterus* in the competitive group would be 5 times higher than that of the competition-absent group. The experimental results showed that the attack frequency of *H. macropterus* in the competitive group at night was about twice that of the competition-absent group, indicating that the presence of shelters reduced the attack frequency of *H. macropterus*. This is consistent with the findings of Chen T. et al. [52]. Wang X. T. et al. found that when the number of corners and the number of crayfish were equal, crayfish survival rates were highest; with fewer corners than crayfish, survival rates were lower; when the number of corners exceeded the number of crayfish, survival rates decreased [53]. The results of this study initially showed that the attack frequency of *H. macropterus* in the 1:1 shelter-to-fish ratio group (competition-absent group) was significantly lower than that in the 1:2 shelter-to-fish ratio group (competitive group). Further research is needed to explore the effects of more detailed shelter number gradients and aquaculture densities on aggressive behavior. Additionally, the experimental results were observed under the condition of placing three types of shelters in the same water area. The effects of individually placing shelters and the presence of other biological interferences on the aggressive and feeding behaviors of *H. macropterus* require further study.

While this study provides novel insights into the aggressive behavior regulation of *H. macropterus*, several limitations should be acknowledged: (1) The experimental fish exhibited weak sexual dimorphism, making sex identification of aggressors challenging in mixed-sex groups; (2) the behavioral quantification system relied solely on visual parameters, lacking integration of multimodal data such as bioelectric signals and stress hormones (e.g., cortisol); (3) the neuroendocrine regulatory pathways underlying aggressive behaviors remain unelucidated. There are future plans to employ sex-specific molecular markers, combined with three-dimensional behavioral trajectory reconstruction, to develop a multimodal dynamic assessment system for aggression intensity. Additionally, we aim to decipher the epigenetic regulatory networks of the hypothalamic–gonadal axis, systematically unravel the mechanisms of aggressive behaviors, and develop precision modulation technologies to inform intensive aquaculture practices.

## 5. Conclusions

In conclusion, this study highlights that *H. macropterus* exhibits pronounced nocturnal aggression, with larger individuals and females displaying higher levels of aggressive behavior. The presence of shelters, particularly tiles and pebbles, significantly reduces aggression, suggesting that environmental enrichment can mitigate conflicts. The establishment of a social hierarchy and the preference for specific shelter types underscore the importance of managing group dynamics and habitat design in aquaculture. These findings provide practical recommendations for improving *H. macropterus* cultivation, such as adopting nocturnal feeding and providing adequate shelters. While size segregation is suggested as a potential strategy to enhance welfare, further experimental validation in *H. macropterus* would strengthen this approach. This research contributes valuable insights into the behavioral ecology of *H. macropterus* and offers strategies to optimize aquaculture practices for them, and similar species.

## Figures and Tables

**Figure 1 animals-15-01192-f001:**
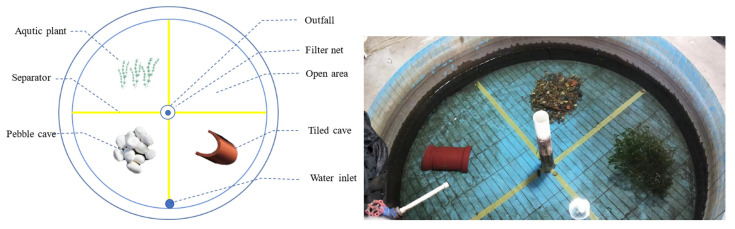
The sketch map and physical map of experiment environment.

**Figure 2 animals-15-01192-f002:**
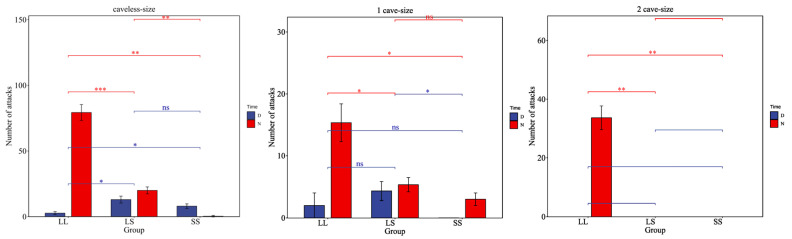
The impact of size on the aggressive behavior of *H. macropterus.* D (daytime, 06:00–19:00), N (nighttime, 19:00–06:00). ns indicates no significant difference (*p* ≥ 0.05), * denotes significant difference (*p* < 0.05), ** represents highly significant difference (*p* < 0.01), and *** represents extremely highly significant difference (*p* < 0.001).

**Figure 3 animals-15-01192-f003:**
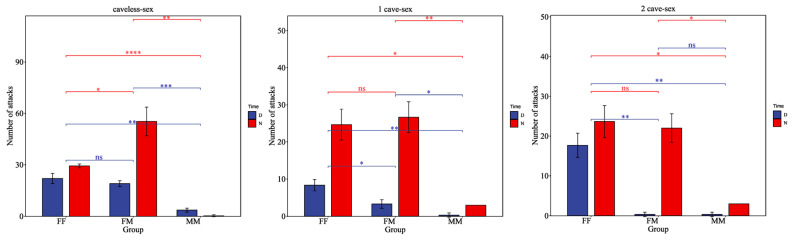
The impact of sex on the aggressive behavior of *H. macropterus.* D (daytime, 06:00–19:00), N (nighttime, 19:00–06:00). ns indicates no significant difference (*p* ≥ 0.05), * denotes significant difference (*p* < 0.05), ** represents highly significant difference (*p* < 0.01), *** represents extremely highly significant difference (*p* < 0.001), and **** represents an extreme significant difference.

**Figure 4 animals-15-01192-f004:**
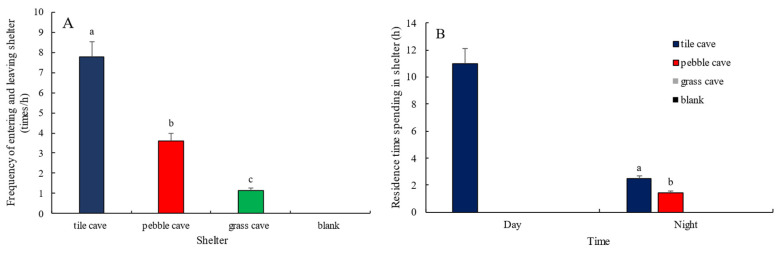
Frequency of *H. macropterus* entering and leaving shelters ((**A**), *n* = 3) and residence time in shelters ((**B**), *n* = 3) in the free-choice group Different lowercases indicate significant difference (*p <* 0.05) among different shelters.

**Figure 5 animals-15-01192-f005:**
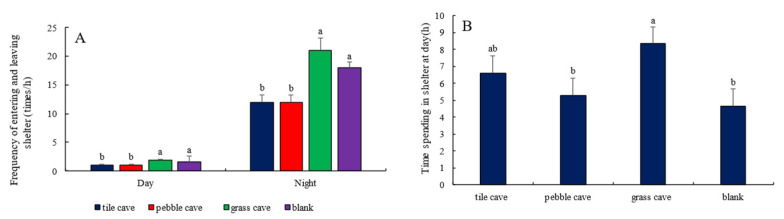
Frequency of *H. macropterus* entering and leaving shelters ((**A**), *n* = 9) and residence time in shelters ((**B**), *n* = 9) in the competition-absent group. Different lowercases indicate significant difference (*p* < 0.05) among different shelters.

**Figure 6 animals-15-01192-f006:**
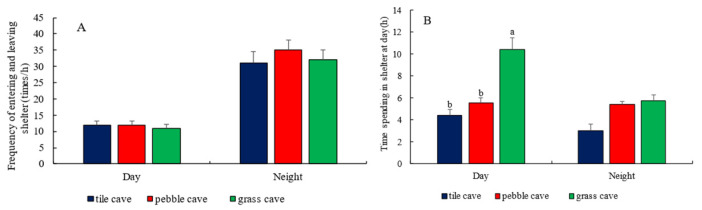
Frequency of *H. macropterus* entering and leaving shelters ((**A**), *n* = 18) and residence time in shelters ((**B**), *n* = 18) in the competitive group. Different lowercases indicate significant difference (*p* < 0.05) among different shelters.

**Figure 7 animals-15-01192-f007:**
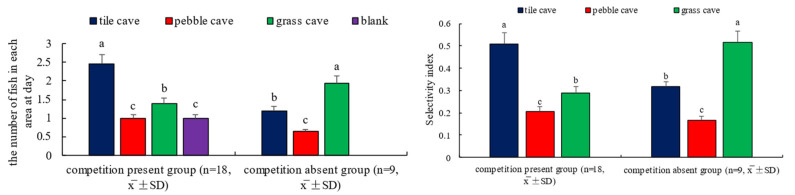
The distribution in various regions and the selectivity index of shelter for *H. macropterus*. Different lowercases indicate significant difference (*p* < 0.05) among different shelters in the same experimental group.

**Table 1 animals-15-01192-t001:** Specifications of *M. macropterus* specimens in this study.

	Weight (g)	Length (cm)	Number of Fish
Experiment 1—Size	164.25 ± 20.29 (large) 45.67 ± 7.67 (small)	27.25 ± 1.44 (large) 16.42 ± 1.32 (small)	18
Experiment 2—Sex	133.69 ± 22.92 (female) 134.83 ± 9.61 (male)	25.48 ± 1.87 (female) 26.27 ± 0.25 (male)	18
Experiment 3—The shelter	74.23 ± 0.90	19.17 ± 0.58	30

**Table 2 animals-15-01192-t002:** Common behaviors and their characteristics of *H. macropterus*.

Types of Behavior	Name of Behavior	Specific Description of Behavioral Characteristics
**Daily Behavior**	Probe	Repeatedly extends its head or part of its body from a hidden cave; swims out a short distance (within the designated hiding area) and quickly returns to the cave; probes towards the water surface. This behavior occurs after the fish is placed in the tank for the first time.
	Explore	Following probing behavior, the fish leaves the cave, swims further distances (crossing partition lines), explores other caves, and investigates the upper water column (swimming in higher water levels).
	Patrol	The fish swims out to inspect the surroundings of the cave and then returns.
	Cruise	The fish swims around the tank without being fixed to any particular spot.
	Rollover	The fish swims with its side facing upwards (commonly observed when entering pebble caves or resting at drainage pipes).
	Predation	When prey fish swims near the cave entrance, the *H. macropterus* strikes quickly and then returns to the cave.
	Thigmotactic	The fish seeks contact with objects (such as drainage pipes or the exterior of tile caves) and remains in contact for an extended period.
**Territorial Behavior**	Invade	An intruder forces the resident fish out of its territory and occupies that territory.
	Drive	The resident fish exhibits active aggression towards the intruder, attempting to drive it away from its territory when the intruder approaches. This includes rapidly charging at the intruder and continuing the pursuit for a certain distance after the intruder attempts to escape.
	Defend	The resident fish actively attacks and pursues any nearby intruders before returning to its original position.
	Guard	The resident fish frequently enters and exits the cave, patrols within the vicinity, and swims a short distance away before returning to the cave.
	Deter	After being driven away, the intruder attempts to approach and re-enter the cave but quickly retreats upon observing the resident, who does not demonstrate aggressive behavior but has established a deterrent effect on the intruder.
**Aggressive Behavior**	Chase	Occurs when Fish A approaches Fish B, resulting in Fish A persistently getting closer while Fish B continuously evades or flees; typically happens outside of territorial boundaries.
	Reverse Chase	Fish B rapidly turns around to pursue Fish A while Fish A is chasing.
	Crash	Two fish collide head-on.
	Bite	Use of the mouth to make contact with the head or body of another fish.
	Active Attack	The fish suddenly accelerates when swimming near a stationary or slowly moving fish, exhibiting chase-like behavior.
	Elude	The fish escapes when under attack.
	Avoid Each Other	After encountering each other, both fish quickly turn and swim away.

## Data Availability

The data that support the findings of this study are available from the first author, X.L.

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
