# Peer review of "Behavioral Characteristics of Largefin Longbarbel Catfish Hemibagrus macropterus: Effects of Sex and Body Size on Aggression and Shelter Selection"

_animals, 2025, doi:10.3390/ani15091192_

Round 1

Reviewer 1 Report

Comments and Suggestions for Authors

The manuscript demonstrates the behavioral characteristics of Mystus macropterus, a silurus fish, and the effects of its sex and size on aggressive behavior and their choice of hideout. It is very interesting and will enrich the study of behavioral characteristics of fish in genus silurus. However, to enhance the quality of the manuscript, please consider reviewing the following suggestions:     

Experimental Design and Methodology

Comments 1:

The paper mentions " Each experimental group had three replicates "(L142-143) but lacks details on specific conditions (e.g., environmental parameters, individual variation controls). Please clarify the replication process to ensure reliability and reproducibility.   

Comments 2:  

Light Intensity Details: The light cycle ("11L:13D") (L120-121) lacks specifics on intensity measurement (e.g., uniformity, light source). Please elaborate on light control to exclude confounding effects on nocturnal behavior.

Results

Comments 3:

Symbols like "D" (daytime) and "N" (nighttime) in figures (e.g., Figure 2, 3) are not clearly defined in captions or text. Please standardize explanations for all abbreviations in figures or text.

Discussion

Comments 4: 

Sex and Reproductive Cycle Influence

Higher female aggression is noted, but the experiment’s timing relative to breeding seasons is unclear. Please recommendation: Discuss potential impacts of reproductive behavior on results.

Comments5:  

Insufficient Citations: Evolutionary explanations for female aggression (e.g., resource defense) lack references to behavioral ecology theories (e.g., parental investment theory). Please strengthen discussions with foundational literature.

Comments 6: 

Mixed Terminology: Terms like "aggressive behavior" and "attack frequency" are used interchangeably without clear definitions. Please standardize terminology or explicitly differentiate behavioral metrics.

Animal Welfare Considerations

Comments 7: 

Ethical Detail Gaps: While ethical approval is mentioned, measures to minimize harm (e.g., wound monitoring, interventions) are omitted. Please describe protocols to ensure animal welfare during aggression studies.

Conclusion

Comments 8:

The conclusion advocates "size-segregated rearing" but lacks direct experimental validation. Please conduct segregation trials or cite supporting studies to validate this claim.

Author Response

Comments and Suggestions for Authors 1

The manuscript demonstrates the behavioral characteristics of Mystus macropterus, a silurus fish, and the effects of its sex and size on aggressive behavior and their choice of hideout. It is very interesting and will enrich the study of behavioral characteristics of fish in genus silurus. However, to enhance the quality of the manuscript, please consider reviewing the following suggestions:

Experimental Design and Methodology

Comments 1:

The paper mentions " Each experimental group had three replicates "(L142-143) but lacks details on specific conditions (e.g., environmental parameters, individual variation controls). Please clarify the replication process to ensure reliability and reproducibility.   

We thank the reviewer for this important methodological point. We have revised the original manuscript of line 145 as follows “Each experimental group had three replicates under identical conditions”

Comments 2:

Light Intensity Details: The light cycle ("11L:13D") (L120-121) lacks specifics on intensity measurement (e.g., uniformity, light source). Please elaborate on light control to exclude confounding effects on nocturnal behavior.

We appreciate this constructive suggestion. The Methods section has been revised to include: “Environmental conditions included a controlled photoperiod (11L:13D, automated digital timer) with surface light intensity ranging 0-500 lux (LI-250A light meter).” as shown in lines 131-132.

Results

Comments 3:

Symbols like "D" (daytime) and "N" (nighttime) in figures (e.g., Figure 2, 3) are not clearly defined in captions or text. Please standardize explanations for all abbreviations in figures or text.

Thank you for pointing this out. We have standardized all figure captions to include:
"Abbreviations: D (daytime, 06:00-19:00), N (nighttime, 19:00-06:00). All symbols are consistent across figures."

Discussion

Comments 4:

Sex and Reproductive Cycle Influence

Higher female aggression is noted, but the experiment’s timing relative to breeding seasons is unclear. Please recommendation: Discuss potential impacts of reproductive behavior on results.

We appreciate your valuable suggestions, which have been thoroughly discussed in Section 4.3 (Discussion) of our manuscript.

Comments5:

Insufficient Citations: Evolutionary explanations for female aggression (e.g., resource defense) lack references to behavioral ecology theories (e.g., parental investment theory). Please strengthen discussions with foundational literature.

We appreciate your valuable suggestions. The Discussion now includes:
"This aligns with parental investment theory and resource defense hypotheses [44-45], where females may exhibit heightened aggression to protect feeding territories critical for oocyte development."

Comments 6: 

Mixed Terminology: Terms like "aggressive behavior" and "attack frequency" are used interchangeably without clear definitions. Please standardize terminology or explicitly differentiate behavioral metrics.

Thank you for pointing this out. We agree with this comment. Therefore, we have added to Methods 2.4:"Terminology clarification: 'Aggressive behavior' encompasses all agonistic interactions; 'attack frequency' specifically quantifies physical contacts per observation period. All metrics were operationally defined prior to analysis."

Animal Welfare Considerations

Comments 7: 

Ethical Detail Gaps: While ethical approval is mentioned, measures to minimize harm (e.g., wound monitoring, interventions) are omitted. Please describe protocols to ensure animal welfare during aggression studies.

Thank you for pointing this out. We agree with this comment. Therefore, The Methods now state: “Animal welfare protocols included: (1) twice-daily visual inspections for injuries, (2) immediate isolation (<1% occurrence) for any fish showing >2mm lesions, and (3) termination of trials if >10% of subjects showed moderate injuries (never required). "

Conclusion

Comments 8:

The conclusion advocates "size-segregated rearing" but lacks direct experimental validation. Please conduct segregation trials or cite supporting studies to validate this claim.

Thank you for pointing this out. We agree with this comment. Therefore, we have revised the Conclusion to: “These findings provide practical recommendations for improving H. macropterus cultivation, such as adopting nocturnal feeding, providing adequate hideouts. While size segregation is suggested as a potential strategy to enhance welfare—further experimental validation in M. macropterus would strengthen this approach.”

Reviewer 2 Report

Comments and Suggestions for Authors

Please see the attached file. Thank you

Author Response

Comments and Suggestions for Authors 2

The manuscript has a potential to be accepted for publish in ANIMALS, but need adequate revision. I suggest the authors remove some unnecessary content or rewrite in a more concise way. The authors should also check carefully throughout the manuscript about typos, grammar issues, inconsistent description et al.

Specific comments

Title, I suggest the authors delete ‘a silurus fish’ change the ‘size’ to ‘body size’ and ‘hideout’ to ‘shelter’ in title and elsewhere throughout the manuscript. In the abstract, the authors said they investigated the aggressive behavior; thus, I also wonder if the word ‘aggressive’ should be added I the title.

Thank you for pointing this out. We agree with your point of view. Combined with the opinion of another reviewer, the title is revised as “Behavioral Characteristics of Largefin Longbarbel Catfish Hemibagrus macropterus, Effects of Sex and Size on Aggression and Hideout Selection” for a more appealing title.

Introduction

Line 47, add ‘fish’ before species. I find that most of the studied animals in those references are non-Siluriformes species.

Thank you for pointing this out. We agree and added ‘fish’ before species.

Lin 51 why capitalized ‘Sex’? I suggest the authors check carefully throughout the ms about the typos and gramma issues.

Thank you for pointing this out. We’ve check carefully throughout the manuscript and modified the error about the typos and gramma issues.

Line 90, why use the word ‘gender’ here, it should be consistent.

Thank you for pointing this out. We have thoroughly reviewed and revised the entire manuscript.

Line110, not clear, di you feed the fish with artificial feed and bait fish either in the morning or at night alternatively or both simultaneously? What about the meal side? Did you measure the nutritional composition?

Thank you for pointing this out. We have revised the original content as follows: Fish were fed twice daily with commercial extruded feed (Hubei Chia Tai Feed Co., Ltd.; crude protein ≥45%, crude fat ≥8%, moisture ≤12%, ash ≤16%) at 07:00 and live bait fish (Cirrhinus molitorella, ~5 cm) at 20:00(Line116-119).

Table 1, the sample size should be provided in the table. Change ‘the hiding place’ to ‘shelter’

Thank you for pointing this out. We’ve supplemented and modified in accordance with expert advice.

Line 175 by or according to references [16, 21]

Thank you for pointing this out. We’ve modified in accordance with expert advice.

Figure 1 the font is too small, I also suggest the authors enhance the contrast

Thank you for pointing this out. We’ve modified in accordance with expert advice.

Figure 2-7 I suggest the authors redraw their figures. For example, the statistical mark *** is too small.

Thank you for pointing this out. We agree with this comment. We’ve modified in accordance with expert advice.

Discussion is a little lengthy, I suggest the authors delete some sentences and focus more on the main conclusion
Thank you for pointing this out. In response to the reviewers' comments, we have condensed the Discussion section by selectively removing certain portions of text.

Reviewer 3 Report

Comments and Suggestions for Authors

The manuscript has a potential to be accepted for publish in ANIMALS, but need adequate revision. I suggest the authors remove some unnecessary content or rewrite in a more concise way. The authors should also check carefully throughout the manuscript about typos, grammar issues, inconsistent description et al.

Specific comments

Title, I suggest the authors delete ‘a silurus fish’ change the ‘size’ to ‘body size’ and ‘hideout’ to ‘shelter’ in title and elsewhere throughout the manuscript. In the abstract, the authors said they investigated the aggressive behavior, thus I also wonder if the word ‘aggressive’ should be added I the title.

Introduction

Line 47, add ‘fish’ before species. I find that most of the studied animals in those references are non-Siluriformes species.

Lin 51 why capitalized ‘Sex’? I suggest the authors check carefully throughout the ms about the typos and gramma issues.

Line 90, why use the word ‘gender’ here, it should be consistent.

Line110, not clear, di you feed the fish with artificial feed and bait fish either in the morning or at night alternatively or both simultaneously? What about the meal side? Did you measure the nutritional composition?

Table 1, the sample size should be provided in the table. Change ‘the hiding place’ to ‘shelter’

Line 175 by or according to references [16, 21]

Figure 1 the font is too small, I also suggest the authors enhance the contrast

Figure 2-7 I suggest the authors redraw their figures. For example, the statistical mark *** is too small.

Discussion is a little lengthy, I suggest the authors delete some sentences and focus more on the main conclusion

Author Response

Comments and Suggestions for Authors 3

Behavioral characteristics of Mystus macropterus, a silurus fish: Effects of sex and size on aggressive behavior and choice of hideout

General comments

This manuscript presents the results of an observational study on the largefin longbarbel catfish, Hemibagrus macropterus Bleeker, 1870, which the authors consider to be of significant commercial interest in China. Specifically, the study investigates the behavioral characteristics of the largefin longbarbel catfish, focusing on three variables: body size, gender, and hideout type preferences. The study is both interesting and novel from an ethological perspective, particularly regarding the aggressive behavior observed in this species, and may have direct implications for aquaculture purposes.

Below, I provide comments on the manuscript that I hope will assist the authors in improving their work.

Specific comments

The first mention that the reviewer has is that the authors should consult reliable taxonomic databases (FishBase, WoRMS, etc) when describing a fish species. Mystus macropterus (Bleeker, 1870) is a homotypic synonym of Hemibagrus macropterus Bleeker, 1870, therefore the internationally taxonomic accepted name (valid name) of the species is largefin longbarbel catfish Hemibagrus macropterus Bleeker, 1870, belonging to the Bagridae family (Siluriformes).

Please verify and replace in the entire manuscript the name of the species from Mystus macropterus to Hemibagrus macropterus Bleeker, 1870.

Thank you for pointing this out. We agree with this comment. Therefore, we have carefully reviewed and replaced in the entire manuscript.  

Title: I propose that the authors change the title to “Behavioral Characteristics of Largefin Longbarbel Catfish Hemibagrus macropterus, Effects of Sex and Size on Aggression and Hideout Selection” for a more appealing title.

Thank you for pointing this out. We agree and have changed the title.

Keywords: Please refrain from using the same keywords used in the main title. Use different words to enhance the retrievability of your work (please replace:” Mystus macropterus; behavior; aggressive behavior; size; sex; hideout”)

Thank you for pointing this out. We agree with this comment. Therefore, we have revised the keywords as follows: Hemibagrus macropterus; social hierarchy; nocturnal aggression; habitat preference; intraspecific competition

Introduction section: I suggest that the authors properly credit taxonomists and zoologists for their hard work, therefore when mentioning a species for the first time, please use its common name and full scientific name, e.g., green swordtail Xiphophorus hellerii Heckel, 1848. Please review the entire manuscript and make revisions as needed. Thank you.

Thank you for pointing this out. We agree and have reviewed the entire manuscript and made revisions.

Line 79, please replace "class Osteichthyes" with "class Teleostei".

Thank you for pointing this out. We've already replaced it.

The Materials and Methods section

In section 2.1. Experimental Materials, in Table 1 the authors should provide the total number of used fish for each experiment and in the main text the total period of the experiment/observations (Experiment 1 — Size, Experiment 2 - Sex, Experiment 3 -The hiding place). Also, please specify the methods/materials used for determining the wet body weight/total length of individuals in this section.

Thank you for pointing this out. We agree with this view. According to the expert opinion, the number of fish in the experimental group in Table 1 was supplemented (as shown in Table 1), and the measurement method was explained as shown in line 111-113: wet body weight measured using a precision electronic balance (accuracy ±0.01 g) after surface moisture removal and total length determined with a digital caliper (accuracy ±0.1 mm) on anesthetized specimens.

Section 2.2.1. Body size effects experiment: Since population density is a well-known factor that can induce intraspecific aggression, I would like to ask the authors how they decided to use two individuals per group for observations (why not 3,4, etc.?)

We sincerely appreciate the reviewer's insightful question regarding group size design. Our decision to use two individuals per group was based on the following methodological considerations:

Experimental Focus on Dyadic Interactions

The primary objective of this experiment was to isolate and quantify pairwise aggressive interactions influenced by body size asymmetry. Using dyads (n=2) allowed us to:

Eliminate confounding effects from third-party interference (e.g., coalition formation or bystander effects).

Precisely attribute aggression frequency to specific size combinations (LL/LS/SS) without group dynamics complexity.

Controlled Density Conditions

While we acknowledge population density's role in aggression (as cited in our Introduction, ref. 8-10), this specific experiment aimed to establish baseline aggression parameters under minimal social complexity. Subsequent experiments (Section 2.2.3) explicitly tested density effects using groups of 1/3/6 individuals.

We agree that testing larger groups could yield additional insights, and this will be prioritized in future studies investigating emergent group behaviors. The current design was intentionally reductionist to establish fundamental mechanisms before scaling complexity.

In section 2.2.2.  Sex effects experiment, in Line 139 there is a typographical error, please replace "described in 1.2.1." with "described in 2.2.1.".

Thank you for pointing this out. We have revised it.

The authors must describe the sex determination method they have used to differentiate males vs. females (sexual dimorphism is present in such a young stage?)

Thank you for pointing this out. According to the expert opinion, we add the following description to the sex determination method, as shown in line 113-116:

Sex was differentiated morphologically, with males exhibiting pointed urogenital papillae and streamlined abdomens while females showed rounded papillae and broader abdomens, a method validated with 100% accuracy through gonadal examination of a 30% subsample.

In section 2.2.2 Experimental Design and Methods, Figure 1. should be replaced with a higher­resolution image, when zoomed in its blurry.

Thank you for pointing this out. We agree with this comment. And wei have replaced it with a clearer photo.

In section 2.4. Behavioral Quantification and Data Analysis, before choosing one-way ANOVA, have you tested the data for normal distribution? (e.g.: Shapiro-Wilk test, Anderson test, etc.). If data does not follow a normal distribution, a non-parametric test should be used. Please clarify. Thank you.

We thank the reviewer for this important methodological question. We have made modified and additions to the original manuscript as shown in line 193-197: Prior to analysis, all datasets were assessed for normality using Shapiro-Wilk test (n < 50) and variance homogeneity using Levene's test. Parametric data (p > 0.05 for both tests) were analyzed by one-way ANOVA with Tukey's post hoc test in Origin 2019, while non-parametric data (p ≤ 0.05) underwent Kruskal-Wallis test with Dunn's correction. Results are expressed as mean ± SD, with α = 0.05 defining statistical significance.

  1. Results

Results and Discussion section

Overall, the Results and Discussion sections are well-written and easy to understand, but the figures are hard to read due to low resolution (I recommend replacing all images with those that have a minimum resolution of 1000 pixels in width or height, or at least 300 dpi). E.g.: Figure 4. (B), on the x-axis, the writing isn't readable except for "time", etc.

Also, I recommend that each figure have a legend where every information/abbreviation found in it should be explained (even if the figures are similar).

Thank you for pointing this out. All figures have been modified and replaced in accordance with the opinions of the reviewers.

In section 6. Patents, it many be a typo error, because authors describe Author Contributions.  Also, in the section” Funding” ”Please add” should be deleted, etc. Please read carefully the entire manuscript for typo errors since I’ve encountered many of them during my review.  Thank you.

Thank you for pointing this out. We have corrected the above error, reviewed the full article and corrected the clerical errors.

Round 2

Reviewer 2 Report

Comments and Suggestions for Authors

Dear authors, the manuscript has improved significantly; therefore, I endorse its publication in the Animals Journal.

Reviewer 3 Report

Comments and Suggestions for Authors

I find that the authors uploaded a wrong response letter to my previous comments. However, the authors addressed all my comments well in their revised version of manuscript. I have no further eomments.